# Qualitative study to develop processes and tools for the assessment and tracking of African institutions' capacity for operational health research

Selina Wallis,[1] Donald C Cole,[2] Oumar Gaye,[3] Blandina T Mmbaga,[4] Victor Mwapasa,[5] Harry Tagbor,[6] Imelda Bates[2]

[1]International health, Liverpool School of Tropical Medicine, Liverpool, UK
[2]Dalla Lana School of Public Health, University of Toronto, Toronto, Canada
[3]Faculty of Medicine, Pharmacy and Dentistry, L'Université Cheikh Anta Diop, Dakar, Senegal
[4]Kilimanjaro Christian Medical University College, Moshi, Tanzania
[5]College of Medicine, Univeristy of Malawi, Malawi
[6]School of Medical Sciences, Kwame Nkrumah University of Science and Technology, Kumasi, Ghana

**Correspondence to**
Prof. Imelda Bates;
imelda.bates@lstmed.ac.uk

## ABSTRACT

**Objectives** Research is key to achieving global development goals. Our objectives were to develop and test an evidence-informed process for assessing health research management and support systems (RMSS) in four African universities and for tracking interventions to address capacity gaps.

**Setting** Four African universities.

**Participants** 83 university staff and students from 11 cadres.

**Intervention/methods** A literature-informed 'benchmark' was developed and used to itemise all components of a university's health RMSS. Data on all components were collected during site visits to four African universities using interview guides, document reviews and facilities observation guides. Gaps in RMSS capacity were identified against the benchmark and institutional action plans developed to remedy gaps. Progress against indicators was tracked over 15 months and common challenges and successes identified.

**Results** Common gaps in operational health research capacity included no accessible research strategy, a lack of research e-tracking capability and inadequate quality checks for proposal submissions and contracts. Feedback indicated that the capacity assessment was comprehensive and generated practical actions, several of which were no-cost. Regular follow-up helped to maintain focus on activities to strengthen health research capacity in the face of challenges.

**Conclusions** Identification of each institutions' strengths and weaknesses against an evidence-informed benchmark enabled them to identify gaps in in their operational health research systems, to develop prioritised action plans, to justify resource requests to fulfil the plans and to track progress in strengthening RMSS. Use of a standard benchmark, approach and tools enabled comparisons across institutions which has accelerated production of evidence about the science of research capacity strengthening. The tools could be used by institutions seeking to understand their strengths and to address gaps in research capacity. Research capacity gaps that were common to several institutions could be a 'smart' investment for governments and health research funders.

## Strengths and limitations of this study

► This study uses qualitative research to generate primary, prospective, longitudinal data about the baseline status of operational health research systems in four African institutions, and tracks changes in research capacity against predetermined indicators.

► The use of the same benchmark and research approach across different institutions enables comparisons to be made so common challenges can be identified; these could be effective targets for investment.

► The main limitations for the study were that the limited follow-up time did not allow for demonstration of the long-term sustainability of changes to research systems and, because our study was designed to provide a broad overview of an institution's research management and support systems (RMSS), it did not explore particular components in depth.

► Institutions found the evaluation process to be comprehensive and helpful since in addition to advancing the science of research capacity strengthening it generated practical actions and progress indicators, and facilitated interinstitutional comparison and benchmarking.

## INTRODUCTION

### Importance of research for development

Health research has been acknowledged to play a key role in progress towards the Sustainable Development Goals.[1] Strong research institutions and skilled researchers are essential for low-and-middle-income countries (LMICs) to generate evidence for their own health policies and to make progress in achieving their health-related goals.[2][3] Investments in operational health research capacity can provide positive returns by promoting evidence-informed policy and practice in the health system,[2] although implementation[4] and estimation of returns can be challenging.[5] The first African Ministerial Conference on

Science and Technology in 2003 recognised that 'there is strong evidence that using research evidence to inform policy and practice leads to benefits which contribute to socioeconomic development'[6] and participating countries committed to spend at least 1% of their gross domestic product on research and development by 2010.[7] Only Kenya, Malawi and South Africa have managed to approach this target and Kenya, Mozambique, Senegal and Uganda all have >40% of their research and development financed from abroad.[7]

### Lack of research/researchers in LMICs especially in Africa

Although the average growth rate of scientific production in Africa is faster than that of the world as a whole, African Union countries only produce 2% of the world's total scientific output.[8] Egypt, Kenya, Nigeria and South Africa produce the largest number of publications from Africa.[7] This is a reflection of the small numbers of researchers in Africa and decades of underinvestment in research institutions. Most countries in sub-Saharan Africa have <500 researchers (of all disciplines) per million inhabitants (eg, Tanzania 35, Ghana 39, Malawi 50, Senegal 361) compared with >4000 per million inhabitants in the UK and North America.[9] There are numerous disincentives to pursuing a research career in many African countries including heavy teaching loads, weak organisational research systems, lack of national research leadership, limited access to scientific information, slow internet connections and inadequate physical facilities including libraries and laboratories.[10]

### Attempts to address weak capacity for operational health research in Africa

Resources to guide development of operational health research capacity have been available for at least a decade[11] but outdated and ineffective models for strengthening capacity persist.[12] African research institutions have historically faced numerous challenges.[13] The ability to produce international quality health research depends not only on developing a critical mass of African researchers, but also on providing them with a conducive environment in which to do research and progress their careers.[14 15] International funders have responded by supporting strengthening of national systems and structures for operational health research[16] and in boosting the capacity of LMIC universities in research governance and management.[17] However, despite long-standing calls for more robust evaluations of capacity development,[11] the evidence needed to inform effective implementation and evaluation of programmes for strengthening operational health research capacity remains weak.[18 19] Furthermore, the lack of clearly defined goals and baselines against which to evaluate the success of research capacity strengthening programmes makes it difficult to track their progress and impact.[20]

Development funders and policymakers are calling for a 'significant re-think of the approach to capacity development'.[21] They stress the need for an interdisciplinary approach which recognises the complexity, fluidity and non-linearity in human systems, a systematic perspective and acknowledgement of relationships between capacity at the individual, institutional and wider societal levels.[19 22] To promote a more purposeful and strategic approach to strengthening operational health research capacity in LMICs, a group of international funders have produced guidance about developing shared principles and indicators,[23] and for evaluating outcomes and impacts of health research capacity strengthening interventions. Putting these guidelines into practice at the organisational level is challenging since little is known about what information matters for strengthening research capacity, and how and why this varies in different institutional contexts.

### Purpose

The purpose of our study was to develop and test an evidence-informed process that could be used (1) to conduct a baseline assessment of operational health research management and support systems (RMSS) in four African universities and (2) to document actions taken to address identified gaps. As institutions implemented these actions, we sought to identify common difficulties they encountered. This information would help not only the institutions, but also external agencies and national governments, to more effectively target and monitor their contributions to strengthening institutional and hence, national health research capacity. The assessment process covered all the components needed for a university to generate, manage and disseminate operational health research of international quality.

### Approach to the study

The study comprised three phases—construction of a benchmark against which to conduct the baseline assessments of institutions' RMSS, development of data collection tools based on the benchmark and collection and analysis of data during visits to the institutions and the follow-up period. Despite earlier work on research management benchmarking,[24] no single document existed which detailed all the systems needed in a university to foster, support and manage international quality operational health research. Hence, it was necessary to develop a comprehensive description of the components of an 'optimal' scenario[19] as a benchmark against which the baseline assessment could be compared.[22 25] We describe the process of using best available evidence to generate this benchmark as a health RMSS list and used the benchmark to craft tools for collecting baseline data in each of the universities and to collate a list of indicators for monitoring progress. We share our experience of using the tools to identify institutions' RMSS capacity gaps, the early results on tracking the universities' progress and challenges in strengthening their RMSS and senior researchers' experience with the RMSS assessment process.

## METHODS

### Partner universities

We worked with four African universities or research institutions which were partners in the Malaria Capacity Development Consortium (MCDC 2008–2015, http://www.mcdconsortium.org/) funded by the Bill and Melinda Gates Foundation and the Wellcome Trust. The MCDC supported African scientists to undertake high-quality malaria research and to enhance the operational health research capacity of their home institutions. In particular, the MCDC aimed to strengthen the capacity of the African universities to provide academic, administrative and financial support to generate health research of international quality despite differences in geography, size and maturity of their research infrastructure.

The institutions were based in Anglophone and Francophone countries in West (two), East (one) and Southern (one) Africa. The entry point for our study into each of the universities was the department (or centre) in which the MCDC's collaborating principal investigator was located. These departments had been established between 1957 and 1991; all had active malaria research programmes and offered postgraduate training. At the time of the study, the universities had between 6000 and 60 000 registered students.

### Generation of a list of RMSS components

In order to conduct a holistic assessment of the African universities' health RMSS, it was necessary to first create a benchmark by identifying all the components and related best practice required for the optimal functioning of such systems.[19] As no single document available detailed all these components, we drafted an initial list of components by itemising all activities that occur within a project cycle and by identifying all the support mechanisms that are required to conceive, generate and monitor research and to ensure that research findings are used to inform national health policies and practices. The list identified search terms (eg, research management, research capacity indicators, institutional benchmarking) which guided the collection of relevant information using internet searches. The search for relevant global publications included academic articles and grey literature such as guidelines and regulations governing research aspects of higher education institutions (online supplementary box 1). We also interrogated websites of agencies relevant for each of the themes, and read their reports and documents and any references included therein and consulted with researchers, grants managers and research finance officers within and beyond our own institutions until no new items emerged and saturation was achieved. We aimed to cover aspects of the institutional capacity needed to provide optimal academic, administrative and financial support for operational health research activities from the perspectives of the dean or principal of the institution, faculty research support staff and researchers at different career stages.

From the literature (online supplementary box 1), we extracted a list of all the items relevant for inclusion in a review of institutional RMSS. To help the development of systematic data collections tools, items on the list were grouped into components which were simultaneously adjusted and expanded to encompass all the aspects of RMSS, with no duplication across components (online supplementary box 2). The 'optimal' scenario for an institutional RMSS was therefore derived by amalgamating all the items identified from the literature search and eliminating any redundancy. In order to ensure comprehensiveness and minimise bias, no assumptions were made about what should be included, no selection criteria were applied to the original list of items and they were drawn together under the eight components without losing any of the items. This list of items therefore represented the description of the 'optimal' scenario (ie, benchmark). The final RMSS components encompassed all the RMSS-relevant items identified in the literature and were

1. Research strategies and policies
2. Institutional support services and infrastructure
3. Supporting funding applications
4. Project management and control
5. Human resource management for research
6. Human resource development for research
7. External promotion of research
8. National research engagement.

### Development of tools for data collection

The most appropriate methods to be used for collecting data on each of the components and their associated items during subsequent visits to the universities were determined.[26] The primary data collection tool was a guide for semistructured interviews with different cadres of university staff, supplemented by a list of facilities to be visited at the institutions (ie, library, information technology suite, laboratories) and a list of documents to be reviewed (ie, strategies, policies, regulations, handbooks).

Inclusion of the entire master list of items for every component in every semistructured interview would have been impractical and inappropriate. Since each interviewee would have knowledge of specific aspects of RMSS in their institution, combinations of questions were selected from an overall suite (online supplementary box 3) to construct focused interview guides for different cadres of interviewees (ie, heads of department/institute deans or principals; senior researchers; staff with research support responsibilities such as administration, finance, human resources, communications, ethics and laboratories). For example, questions for laboratory technicians, but not for other cadres, dealt with equipment maintenance. We ensured that all items from the master list were covered across the set of cadre-specific interview guides.

The data collection tools (lists and interview guides) were reviewed by all members of the research team and adjustments were made to reduce redundancy. Additional changes were made after the first university visit and minor revisions were made during the visit to the second

university. After this, no more revisions were required, so this version was used for the two subsequent visits.

## Baseline data collection during university visits

Previsit briefings were conducted by Skype with the MCDC principal investigator in each of the African universities, to explain the purpose and process of the visits and to schedule interviews with different cadres of staff and students. The principal investigators were provided with the data collection tools in advance of the visits so they were aware of the range and type of information that would be sought. Subsequently, 3–4-day visits to each of the four African universities were conducted by 2–3 members of the research team between September and November 2014.

As far as possible, all data collected during the visits were obtained from at least two independent sources to enhance validity.[27] Interviewees were asked if any aspects of research systems had not been covered by the interview questions and, as a result, procurement procedures were added to the questions for the second and subsequent visits. During each interview, interviewees were asked to propose feasible actions that could be taken to overcome any of the challenges or gaps in research support systems that they mentioned.

Notes from the interviews were typed up within a few hours of each interview, checked against audio-recordings of the interviews (available if interviewees gave permission) and final versions were verified among the site visit team. Information from observation of facilities and review of documents was used to elaborate and verify data from the interviews. A consultation meeting was held at the end of each visit for all available interviewees to share preliminary findings about strengths and gaps identified in the institutional RMSS. In keeping with the principles of interdisciplinary team reflexivity[28] and of pooling internal and external assessments,[29] we used the meetings to check the accuracy of the findings, to discuss the reasons for discrepancies, to generate and prioritise proposed actions and to ensure that such actions were deemed feasible by institution staff.

## Baseline data analysis

A framework analysis approach was used to manage and analyse the multidisciplinary information generated from the site visits about institutions' 'baseline' research systems.[30] Data were entered into a matrix which had a row for each of the eight components. Columns for topics within each of the RMSS components that emerged from the interviews were constructed using deductive (ie, based on the topics/items grouped under each component from the scoping review) and inductive (ie, unexpected new topics that emerged from the information collected) approaches. Use of the matrix facilitated identification of emerging patterns and comparison of the strengths and weaknesses in each institution's research systems. Following the site visits, findings were presented in a draft report which was reviewed by the MCDC principal investigators in consultation with

their institutional colleagues, before being finalised. To respect confidentiality, the final reports were only shared with the MCDC secretariat and the institutions themselves. An anonymised 'overview' report was produced and made publicly available which summarised commonalities and differences in RMSS across all institutions and highlighted innovative RMSS practices.[31]

## Follow-up interviews for tracking progress and obtaining feedback on the process

Information about progress and challenges in addressing gaps in the institutions' health RMSS was obtained through 2–5 Skype and telephone interviews with the MCDC principal investigators in each institution over 15 months until May 2016. Each interview lasted 20–40 min and covered the gaps and actions identified in the relevant intuitional baseline report. The relevant principal investigator, in discussion with SW and IB, gauged the progress on each action, explained the means by which progress had been achieved and described any challenges experienced. During the interviews, the principal investigators were asked to comment on whether the process had been helpful, and if so how and which aspects could be improved in the future and to reflect on their role as research manager practitioners.[32] These comments were organised into themes, and quotes reflective of each theme were selected to convey the principal investigators' perspectives in their own words.

Information obtained about progress and challenges around actions in the baseline report was mapped against the eight RMSS components using a pre-prepared matrix and analysed using a framework analysis approach. Two authors (SW, IB) reviewed the self-reported progress of each institution and broadly assessed whether the institutions collectively had made 'good', 'moderate' or 'little/no' progress in addressing the gaps in each component of their research support systems. This helped in understanding which components of research support systems all four universities found most easy to address and which they found hardest. A report outlining progress and challenges was drafted for each institution and reviewed by each principal investigator.

## Ethical considerations

This project was considered to be primarily an evaluation which aimed to improve practices for strengthening research capacity so formal ethical approval was not sought. However, we explained the study to all participants, asked each interviewee for their verbal consent to participate and provided an opportunity for them to refuse without any consequences for themselves.

## RESULTS
### Baseline situation

In total, 83 interviews were conducted (19–22/university) with 11 different cadres of interviewees (table 1), 65 documents/resources (12–20/university) were reviewed,

**Table 1** Number of interviewees for baseline data collection by cadre and institution

|  | Institution number | | | |
|---|---|---|---|---|
|  | 1 | 2 | 3 | 4 |
| Principal investigator | 1 | 1 | 1 | 1 |
| PhD students | 5 | 1 | 3 | 0 |
| Senior lecturer/lecturer/postdoc | 3 | 1 | 2 | 2 |
| Dean/registrar/provost/principal | 4 | 4 | 3 | 5 |
| Head of department | 1 | 3 | 1 | 1 |
| Research/ethics manager/administrator | 2 | 4 | 1 | 2 |
| Human resources staff | 1 | 1 | 0 | 2 |
| Finance/procurement staff | 2 | 2 | 2 | 1 |
| Information technology/library staff | 3 | 3 | 4 | 4 |
| Laboratory staff | 0 | 2 | 2 | 1 |
| Communications staff | 0 | 0 | 1 | 0 |
| Total | 22 | 22 | 20 | 19 |

and facilities observed included libraries, research laboratories and study spaces. The gaps in RMSS that were common (ie, occurred in at least three of the four universities), and proposed actions that emerged during the on-site visits to address these gaps, were categorised by RMSS component (table 2).

### Progress in strengthening universities' RMSS

All of the universities had made some progress in addressing gaps in their research support systems, and there were some common successes and challenges. Examples are provided in table 3. Although the MCDC provided some institutions with limited funding to address some of these gaps, many of the actions, such as reorganisation of management structures or in-house training, did not require additional funds.

Overall, little or no progress was made in Research strategies and policies, External promotion of research and National research engagement; moderate progress was made in Institutional support services and infrastructure and Human resource management and development for research; and good progress in Supporting funding applications and Project management and control. Examples of innovative practices and problem-solving were identified for each component (table 3).

### The process of assessing and tracking strengthening of RMSS

The process of assessing and providing feedback on institutional RMSS used in the study was universally viewed as a positive and constructive way to raise awareness of the importance of strengthening research support systems and to catalyse broader institutional engagement with these topics. Relevant comments from interviews with the principal investigators included:

> Senior staff are really engaging with this. They understand the importance of the programme.

The project definitely helped to raise awareness of all the challenges we are facing, that we need more funds and to improve the environment; it highlighted difficulties and that all the partners are now really interested in helping African institutions. It enabled us to start some concrete actions and now we have institutional buy in, now they are engaged and committed to go further.

An area for improvement was in ensuring that important documents provided to institutions, such as drafts of the research capacity assessments, were produced in French as well as English language.

> It would help if the report was in French, with logos, stamp and signature—an official version. Otherwise a translation is not taken seriously.

The comprehensive nature of the assessments and data collection tools provided confidence that all key aspects of research support systems had been covered during the process and helped stakeholders to prioritise and justify their future budgeting and funding requests.

> It was very useful to get an overview of the whole system from an outside team.

> A piecemeal approach would not be effective at all. We need to look at each area. We can then leverage funding … and use this [assessment] to make sure every area is funded.

The collaboration between an external team and stakeholders within the institutions brought additional benefits in terms of impartiality and reduction in bias, which would not have been possible with an exclusively internal review team. Seeking opinions from multiple perspectives and the involvement of external team helped to overcome internal sensitivities.

> It stimulated honest and fair discussion between us all … It demonstrated our strengths as well as weaknesses. Everyone said it didn't say anything we didn't know but as an outside organisation produced it there were no biases. That's why everyone has agreed we need to move forward.

> Certain areas in the overall report helped when I was presenting the sensitive issues. There are common problems - instead of feeling hopeless, we felt we were doing better [than other institutions] in some areas. We knew … that here are political issues. If the recommendation had come from within that could have caused issues.

Addressing gaps in research support systems is a complex undertaking and regular contact with the external team to track progress was helpful for keeping the focus on priorities and maintaining momentum.

> The follow up process was helpful to keep me focused on understanding the changes occurring across the college and in all areas of research management.

**Table 2** Consolidated key gaps in research management and support systems (RMSS) and proposed actions, by RMSS component

| Gaps | Proposed actions |
|---|---|
| *Research strategies and policies* | |
| ► No research strategy or not available or publicised<br>► Lack of central tracking of research activities | ► Departments/universities need an accessible research strategy with polices and guidelines to support its implementation<br>► Electronic research management support systems are needed to track proposals and projects and to document research income and disbursement including overheads |
| *Institutional support services and infrastructure* | |
| ► Lack of research support offices and/or insufficient coordination between departments and university levels<br>► Inadequate resourcing and lack of clarity about the role and long-term financial sustainability of research support offices<br>► Research laboratory facilities are not accredited and lack overarching planning to harmonise equipment purchase and maintenance across multiple short-term projects<br>► Unclear relationship between hard copy library facilities and increasing use of e-resources | ► The roles and relationships between university-level research coordination and research support offices at faculty or college level need to be clarified<br>► The strategy for research support offices at faculty or college level needs to be clarified and mechanisms found for long-term sustainability and buy-in by the researchers<br>► Achieve international laboratory accreditation for the institution's own laboratories; harmonise research laboratories' activities with those of affiliated organisations and establish clear processes and costs for researchers wishing to access these facilities<br>► Proactively plan the future of book libraries in the context of the shift to increasing use of e-resources, including their possible integration with information and communication technology (ICT) facilities |
| *Supporting funding applications* | |
| ► Insufficient quality assurance checks and signing off processes for proposal submissions or contracts which could put the institution at risk of contractual or intellectual property issues | ► Set up mechanisms for timely, multidisciplinary (eg, finance, legal, ICT, laboratory, library, procurement) input into proposal development<br>► Set up a formal process for quality assurance and authorisation of proposals before submission and for tracking the outcome of submissions |
| *Project management and control* | |
| ► Senior researchers spend a substantial proportion of time on administrative, procurement and other issues that could be more effectively taken on by non-academic professional staff<br>► Lack of systems for tracking financial spend against budget for projects risks underspend or overspend<br>► Unclear lines of responsibility between researchers and finance officers regarding financial tracking and reporting | ► Establish an electronic research information system to systematically manage and track all aspects each project including the project agreement, protocol, budgets, funding requirements, accounting and audit, and to maximise recoupment of overheads<br>► Establish a formal project approval process for successful applications, including and contract review and sign off<br>► Encourage researchers to include and budget for experienced administrators to help reduce the time they spend on project administration and to actively include other relevant inputs such as procurement expertise<br>► Provide joint training in financial management for researchers and finance officers and increase clarity and understanding about their various roles and responsibilities in relation to each other, the institution and the research funders |
| *Human resource management for research* | |
| ► Lack of clarity on contractual arrangements, and therefore institutional responsibility, for short-term project staff<br>► Poorly defined, or non-existent, career paths for non-academic professionals such as ICT, library and administrative staff<br>► No formal postdoctoral career posts for researchers | ► Strengthen human resource skills and structures so that they can better support researchers and research projects, and to ensure that project staff are university employees with access to the protection and facilities of the institution where this is not currently the case<br>► Formalise career tracks for research support staff<br>► Formal postdoctoral training programmes need to be established to develop and retain talented researchers |
| *Human resource development for research* | |
| ► No coordinated, institutionalised programmes for induction or research skills training for researchers; reliance on projects to provide training means focus is on technical skills rather than generic skills, such as leadership and research communication<br>► Training offered at university level (eg, computer skills, literature searching) poorly publicised and used by researchers | ► Provide a formal induction programme and training needs assessment for new research staff<br>► Establish an institutional programme of skills training for researchers, possibly through a dedicated unit, that includes non-technical skills such as leadership, supervision and project management<br>► Improve incorporation of existing training opportunities (eg, provided by library and ICT staff) into a core skills training programme for researchers |
| *External promotion of research* | |

**Table 2** Continued

| Gaps | Proposed actions |
|---|---|
| ▶ Promotion of research activities and successes by the universities not prioritised although widely recognised as important | ▶ Review research section of universities' website to ensure information is current and that hyperlinks are working<br>▶ Consider setting up a unit specifically to enhance the visibility of institutional and/or departmental research activities and outputs<br>▶ Provide training in research communication to improve researchers' ability to write 'jargon-free' communications such as press releases and policy briefs |
| *National research engagement* | |
| ▶ Insufficient publicising of institutional research outputs in influencing national and international policymaking and programming | ▶ Explore options for improving researchers' ability to impact on national health research priorities and practices<br>▶ Universities and departments should systematically document and showcase national and international uptake and use of the research findings they have generated |

On-going follow up was helpful to keep on track with forward movement.

## DISCUSSION
### Process and tools
We have demonstrated that it is possible to construct and implement a coherent, evidence-informed process for assessing and tracking programmes to strengthen institutions' health RMSS. The comprehensive data collection tools drew on current approaches and evidence from several disciplines including research management, education and organisational systems.[33–35] It has parallels with others' efforts[36] to construct assessment tools to improve the quality of indicators and processes for measuring operational health research capacity strengthening.[20] The assessment process was systematic yet flexible enough to accommodate the complexity and fluidity of health RMSS, across a range of African universities. The assessment process acknowledged the influence of inter-relationships between individual, institutional and wider societal levels on the 'research ecosystem' (ie, researchers and their institutions, funders and governments who support research, policymakers who use research and communication specialists who share and discuss the findings with a broad audience).[37] The way in which the assessment process was conducted, particularly the findings from the baseline assessments and the collaborative identification of actions to address health RMSS gaps, was universally viewed as positive and is consistent with others' experience in reviewing operational health research capacity.[4 36] In addition, the institutional assessments helped to raise awareness of the importance of strengthening RMSS[18] and to catalyse multidisciplinary engagement in improving RMSS across the institutions.[38]

Such assessments would be difficult for exclusively internal teams to undertake since they may struggle to gain timely access to senior university officials and could be influenced by sensitivities and politics within the institutions. A partnership between senior institutional researchers, who intimately understood the structural, financial and political context, and an external team, who were impartial and experienced in such assessments, was

therefore essential to maximise assessment validity and contribution to learning.[18] Such insider–outsider assessments have also been used in examining research ethics systems.[29] The transferability of the RMSS assessment tools and processes across geopolitical and institutional boundaries means that they could be usefully deployed in the increasingly common model of research consortia.[24] Of note is the need to produce reports for non-Anglophone universities in the country's dominant language since language barriers are known to be a critical handicap in scientific collaborations and in engaging senior university officials.[39]

### Tracking progress/challenges
Although there are numerous publications of retrospective evaluations of research capacity strengthening efforts, prospective tracking of progress is far less common.[40] We applied an established five-step process for assessing baseline status and prospectively tracking changes in operational health research capacity.[18] The researchers perceived the process as constructive since it helped to maintain focus and momentum within the institution, and provided an opportunity to introduce and share innovative approaches to problem-solving at each institution and for each RMSS component. Most institutions had made the best progress in areas that were primarily under the control of the collaborating senior researchers' departments, such as involving finance officers and managers in developing research proposals, and providing training and resources for managing grants. Much of this progress was achieved with limited or no additional funds. This may therefore be a useful indicator of what might be achieved by other research institutions in Africa who have minimal external support.

Gaps in operational health research capacity that were generally found to be most the challenging to remedy depended on university-wide changes. Examples included embedding research training, which was usually non-sustainably linked to projects, within university systems, and ensuring laboratories were accredited and underpinned by sustainable financing models. Most challenging of all were the lack of systems for communication and dissemination of research outputs and for using research to

**Table 3** Examples of progress on actions, challenges and innovative problem-solving during 2015–2016, by research management and support system component and institution

| Compoent (overall progress) | Institution 1 | Institution 2 | Institution 3 | Institution 4 | Examples of innovative practices and problem-solving |
|---|---|---|---|---|---|
| Research strategies and policies (little/no progress) | University-level strategy being approved but college level not yet developed. A new university-wide research database now in place. A small university research fund has been set up. | Consultation on the strategy has taken place. A draft document is being written to submit for management approval. New research management software has been installed to manage the research database. | A research grant office is planned for early 2016 and a research database will be created within this office. | Strategy is on hold as wide-scale institutional constitutional change is under way. (Post-project note: *Strategic plan 2017–2022 was approved December 2016*.) | ▲ Through a consultative process and workshop involving researchers, university officials, research support staff and representatives from external national agencies, the Ministry of Health and international research consortia, a research strategy was agreed based on a strengths, weaknesses, opportunities and threats analysis. |
| Institutional support services and infrastructure (moderate progress) | Work is ongoing to coordinate inputs to proposals and grant management by principal investigators (PIs) and finance staff. Workshops took place in 2015. | The first Memoramdum of Understanding (MoU) and standard operating procedures (SOPs) are in development for grant-related activities. ICT policies for marketing and staff engagement are being developed. Sustainability planning for lab resources and budgeting is under way. | A new high-speed internet (DSL) has been installed at the Department of Biology, the PhD-dedicated space and the faculty library. Steps towards formal accreditation of quality assurance processes and certification for the department labs have begun. | Progress has been made in establishing a research support office (grants unit) with administrators and financial staff appointed. | ▲ A Laboratory Committee has been formed and inventoried all key laboratory resources within the college and affiliated research institutions. |

Continued

**Table 3** Continued

| Compoent (overall progress) | Institution 1 | Institution 2 | Institution 3 | Institution 4 | Examples of innovative practices and problem-solving |
|---|---|---|---|---|---|
| Supporting funding applications (good progress) | A university-wide database has been implemented. The finance office is now involved in proposal development and joint training has taken place. | A new system for disseminating information about the research support services has been instigated. 'How-to' guideline on developing research proposals with a budget framework and checklists has been launched. A research careers development webpage has been developed by the institution with materials for fledgling researchers. | Implementation of a new research grant office and recruitment of a research coordinator has been agreed. | Administrators from the research/grants office have started assisting the PIs in proposal development, registering projects and tracking implementation and management. The Grant office has been newly registered with several 'calls application portals' and a database of researcher interests is being created. | ▶ Production of a new 'how-to' flow chart outlining the steps needed to develop research proposals. ▶ A framework for calculating a proportion of staff time to be included in proposals for grants management has been established. ▶ Topic-specific research groups have been formed which can respond to funding calls and collaboration opportunities. ▶ A legal unit has been set up to guide the contract review process and a contract template and checklist developed. |
| Project management and control (good progress) | Contract review, sign off and risk management procedures have been strengthened with contracts signed off by provosts or deans of colleges. Need for in-house training for admin to support PIs identified. | New systems for pre-award and post-award management have been developed. A new policy for compulsory post-award 'Grants Management' inductions for all successful principal investigators is available. A grants administrator has now been appointed. A new legal unit has been set up as part of research support services. | SOPs for expenses and account management have been developed and validated with PIs, finance and administrative officers. | Project management procedures are now overseen by an administrator who manages the grant and liaises with the PI on the implementation of the project. A new financial management system PASTEL is now being used but staff need training to upload project data. | ▶ Standard operating procedures for expenses and account management have been developed jointly by researchers, finance and administrative officers. ▶ New quarterly meetings between finance officers and researchers are used to address financial management of project grants. |

Continued

**Table 3** Continued

| Compoent (overall progress) | Institution 1 | Institution 2 | Institution 3 | Institution 4 | Examples of innovative practices and problem-solving |
|---|---|---|---|---|---|
| Human resource management for research (moderate progress) | Human resource recruitment policies and procedures for project staff have been reviewed and developed. A new staff development programme has been launched. Ongoing meetings are continuing to discuss the need to develop a postgraduate programme. | A new training needs assessment for research support staff has been completed and training plans developed. Procedures for hiring post doctoral fellows have been established and terms and conditions of service defined. | The introduction of project management training courses and cross-site visits for administrative staff and early career researchers have strengthened human resource management. Staff needs for personal development have been identified but there is yet to be a plan to address these. | A team has been formed to develop a research scale based on a national scale for researchers. Postdoctoral programmes are not yet institutionalised. | ▶ Career Development Centres have been established and stocked with training resources. ▶ Human resources staff are being included in planning for research staff development. |
| Human resource development for research (moderate progress) | Work is still ongoing to plan a formal research-based induction process for project staff and fledgling researchers. Mentorship training and pairing has taken place. | Work has begun towards a comprehensive competency-based capacity building programme for junior and senior researchers. A research capacity gap validation workshop and a training needs assessment for research staff have been conducted. A new annual research training calendar has been developed. | Progress made towards adaptation and institutionalisation of the doctoral research courses for researchers. An external funder is working with the institution to plan personal development and leadership courses. | The role of human resource teams in research staff development has been reviewed and defined. More work needs to be done to build a structured programme of capacity development for new and existing staff in research skills. | ▶ An annual research training calendar has been developed for research staff based on a formal training needs assessment. ▶ Regular training is now provided by the university on the roles and responsibilities of research leaders, administrators and finance staff throughout the grant life cycle. |

Continued

**Table 3** Continued

| Compoent (overall progress) | Institution 1 | Institution 2 | Institution 3 | Institution 4 | Examples of innovative practices and problem-solving |
|---|---|---|---|---|---|
| External promotion of research (little/no progress) | Limited training and mentoring in research communication provided by an external agency has started. Some improvement of the research projects on the college website but more needs to be done to encourage researchers to add their details. The academic board are to incorporate research uptake and the management of research outputs, into the university's centralised research policy. | Research profiles of faculty are being introduced onto their webpages. A separate research office website is under construction. There are plans for a Knowledge Management Unit to help achieve better visibility of research activities. | The Faculty of Medicine website has been updated with current research projects. The recruitment of a communication officer is planned, who will be responsible for supporting the publicising and dissemination of research activities and uptake. | The institution has an ongoing strong external collaborative network including an annual PhD symposium. The planned new research strategy will embed research dissemination and uptake as a high-priority area. Work is ongoing to restructure the website. | ▶ An attractive annual university research report which chronicles recent research activities and highlights individuals, departments and colleges has been produced and is publicly available. |
| National research engagement (little/no progress) | Mechanisms still need to be established at college and departmental level to promote, monitor and record communication and to share opportunities for engagement between projects. Continuation of institution's work with multinational partners on national research engagement to feed into the college's research strategy. | Key national health policymaking forums have been targeted with specifically designed materials. A local travel grant system is being established to support researchers to attend national meetings. A template has been developed for annual research reports, to highlight successes, failures and opportunities for policy engagement. | Ongoing participation in national tailored monitoring and evaluation training courses for medical district officers and health supervisors where research results and experience will be shared. | Researchers continue to attend a national research conference (eg, in 2015) to disseminate their projects. Promotion of their research into policy and practice is not yet stated in the research strategy as a core element of a researchers' responsibility. | ▶ A university research uptake strategy has been published. ▶ University regularly hosts a meeting of leading thinkers in science, policy, industry and civil society in Africa. |

influence health policies and programmes. This lack of institutional knowledge exchange capacity to promote research uptake in Africa has been noted by others.[41]

## Limitations of the study

Our study was designed to provide a broad overview of an institution's health RMSS, and therefore could did not explore particular components in depth. Other instruments and guidelines are available to do this including Good Financial Grants Practice,[42] for researchers' development framework,[34] Octagon for research ethics capacity,[29] 'stepwise' laboratory accreditation[43] and Development Research Uptake in Sub-Saharan Africa (DRUSSA) for research uptake.[44] The MCDC principal investigators varied in their seniority, influence and social capital[45] (ie, the norms and networks that enable people to act collectively) which may have affected the thoroughness of the assessment phase, as well as the extent of progress especially in implementing university-wide actions. We recognise that the study only included four African institutions and that these cannot be considered representative of the diversity and complexity of universities within the continent and even within individual countries. The lack of a theory of change[46] for the broader MCDC programme meant that explicit articulation of a common set of outcomes and pathway to change for strengthening RMSS was lacking.[47] Tracking information in progress was generally not independently verified as it was based on Skype or phone interviews with the MCDC principal investigators. The follow-up time was 15 months which is too short to be able to demonstrate longer-term impact of such a process on health RMSS. Hence, we regard our prospective tracking as an initial experience which could be used to guide a more fulsome, prospective evaluation.

## Contributions to an emerging science

Momentum is gathering around a new global science on research capacity strengthening which draws on implementation research,[48] research evaluation processes[5] and qualitative research methodologies.[49] Our effort is consonant with this developing global science, addressing the area of health RMSS with an explicit and comprehensive set of assessment tools, embedded in a collegial, collaborative process. Similar to a small but growing number of colleagues engaged in contributing to the science-base for research capacity strengthening, we are sharing our tools in a peer-review forum, so that others can apply and adapt them for assessing their own or others' university's RMSS. Linking collaborative RMSS assessments of gaps with collegial generation of actions to address those gaps, and jointly tracking progress on chosen actions and challenges prospectively constitutes a more rigorous approach to operational health research capacity strengthening than has been common to date.[20] In addition, documentation of innovative problem-solving by African institutions is crucial to counter deficit-focused narratives, facilitate sharing among resource-constrained institutions and facilitate universities' role as agents of change.[50] An additional benefit of using a systematic, common approach to strengthening institutional health research capacity is that it provides evidence for external agencies and governments about better targeting of efforts to make institutions in Africa globally competitive research leaders.

## Implications

Research capacity outputs need to be recognised as of equivalent value to research outputs[12] and therefore need a rigorous scientific basis. Our experience in developing and applying an assessment and tracking framework can facilitate similar initiatives in other research-oriented institutions in LMICs and their respective consortia. The identification and sharing of RMSS components that are commonly problematic could guide national governments to target their resources towards these weakest components. At the supranational level, the use of our tools and process, and sharing of the results more widely, enables comparisons to be made across institutions and countries. Such analyses would not only contribute to the science of operational health research capacity strengthening, by enabling common research approaches and tools to be applied in different contexts and by validating findings on common capacity gaps, but also provide guidance to international health and research funders about 'smart' investment of resources. Sharing of problem-solving innovations in RMSS among universities and research institutes with similar resource constraints through such organisations as the African Academy of Sciences is an important more immediate opportunity. Finding ways to share such innovations widely beyond health, for example, through interdisciplinary study tours or joint workshops for researchers and research support staff, is imperative for fostering collaborations for RMSS strengthening, and hence health system strengthening more broadly.

**Acknowledgements** The authors acknowledge the collaboration of staff of participating universities, which have remained anonymous for reasons of confidentiality of the findings. Thanks to other members of the study and administration team, Ema Kelly and Vicki Doyle of Capacity Development International, Dorte Holler Johansen from the University of Copenhagen, Denmark, and Lorelei Silvester and Denise Wellings from LSTM for their valuable contributions to this study.

**Contributors** SW and DCC contributed to literature review, study design, baseline and follow-up data collection, analysis and final paper. OG, BM, VM and HT contributed to providing baseline and follow-up data and final paper. IB led study design, contributed to literature review, baseline and follow up data collection, analysis and led drafts of final paper.

**Funding** This work was supported by a grant from the Wellcome Trust, UK, to the London School of Hygiene and Tropical Medicine MCDC project (http://www.mcdconsortium.org/phd-programme.php).

**Disclaimer** The funders had no role in study design, data collection and analysis, decision to publish or preparation of the manuscript.

**Competing interests** None declared.

**Provenance and peer review** Not commissioned; externally peer reviewed.

**Data sharing statement** No additional data are available. All data related to this study are included in this submission, either in tables in the manuscript or in supplementary files.

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
