## [Reviewer comments · BMJ Open]

ARTICLE DETAILS

TITLE (PROVISIONAL)	A qualitative study to develop processes and tools for the assessment and tracking of African institutions' capacity for operational health research
AUTHORS	Wallis, Selina; Cole, Donald; Gaye, Oumar; Mmbaga, Blandina; Mwapasa, Victor; Tagbor, Harry; Bates, Imelda

VERSION 1 - REVIEW

REVIEWER	Dossou Jean-Paul Research Centre in Human Reproduction and Demography, Benin
REVIEW RETURNED	05-Apr-2017

GENERAL COMMENTS	Authors address in this manuscript a relevant subject. This study can contribute significantly to the knowledge on health research capacity strengthening. Here are few comments. 1- Is the abstract accurate, balanced and complete? The conclusion of the abstract presents some result points. For instance « Feedback indicated that the process was comprehensive and generated practical actions, several of which were no-cost. » , belongs to the results paragraph rather than to the conclusion. I encourage the authors to review the whole abstract, to check similar inconsistencies. This should help to have a more balanced abstract. 2- Are the methods described sufficiently to allow the study to be repeated? In the current state of the manuscript, the methods are not appropriately reported. The study design should be clearly defined and justified with regard to the study purposes; thus the link with the data collect technics, tools and analysis can be better understood. Authors conducted a literature review by combining different procedures including internet search. They missed to present useful details about this review. For instance for the internet search : what databases have been reviewed ? when ? which key-words ? What were the selection criteria etc. Other questions include: who developed the tools ? who collected the baseline data ? who collected the follow-up data ? Same or different researchers ? What were their background ? 3- Are research ethics (e.g. participant consent, ethics approval) addressed appropriately? The manuscript do not present the ethical considerations related to this research. 4- Are the results presented clearly? The structure of the results should be linked to the study design, that
--

	seems to be a multiple case study design. If this suggested design is accepted by authors, the results will gain to identify similarities and differences rather than providing only a listing of separate findings in each case (institution). Beyond what can be learned in each cases, authors should analyse what can be learned across cases. This reference may help : Yin R: Case study research. Design and methods. Los Angeles: Sage Publications; 2009. 5- Are the study limitations discussed adequately? Overall the study focuses on gaps and ignore explicitation of existing capacities, and existing good practices. The paper focuses also on the meso-level (academic organization). Whoever several meso-level factors are obviously influenced by the macro-level factors like the National Health Research System (Tikki P, Ritu s, Sadana SH, Zulfiqar AB, A.H. A, Jonathon S. Knowledge for better health — a conceptual framework and foundation for health research systems. Bulletin of World Health Organization. 2003;81:6). The discussion may gain to discuss this interlinks and the choice made by the authors.
--	--

REVIEWER	Nasreen Jessani Johns Hopkins University - USA Stellenbosch University - S Africa
REVIEW RETURNED	05-Apr-2017

GENERAL COMMENTS	Review: Process and tools for the assessment and tracking of African institutions' capacity for health research Overview: This is a really important paper and a great contribution to learning and practice in this critical area of research capacity development. The authors have done a great job of articulating the justification as well as process of the intervention. The tools could (and should) be used by institutions seeking to better understand their strengths and gaps in research capacity and build upon that for improvement. Overall comments:  - The paper refers to health research in a more general term throughout the document but it would seem like the authors are referring particularly to policy relevant or operational health research (as opposed to medical health research). If this is indeed the correct interpretation then it would be helpful to make this more explicit. - The identity of the 4 institutions can be surmised from the author contributions. Given that the paper would be stronger if the list of institutions and countries was included in the intro and methods sections of the paper while maintaining anonymity in the results (see comments in the abstract, intro and methods section). This would provide a better sense of context (4 african countries out of 54 underappreciates the diversity and complexity within the continent (and oftentimes
--

within a country too).

- The details on how “progress” is being evaluated is extremely vague. The nebulousness creates confusion and frustration at various areas of the paper and it would be helpful to have this articulated with a bit more detail and with the pre-post status of these institutions with respect to the various indicators (see comments in results section)

Specific comments:

Abstract:

- There seems to be a background section missing...This would perhaps link to the find sentence in the conclusions about investing in research capacity.
- It would be helpful to break up the participants into how many were staff and how many were students.
- Were the 4 African universities all in one country? In East and West Africa? South? While it may be important to maintain anonymity of the unis more about their context would be helpful under Setting or include it in Intervention/methods.
- Mixed-methods research often implies quant as well as qual methods. Might be more informative for the reader if the authors indicated “qualitative methods” and then listed the three.
- “progress under indicators was tracked” implied a timeframe that is not included in the abstract. This would be helpful too.

Introduction

- Line 20: “Few African countries” is vague. Would be helpful to know how many and which ones.
- Line 32: It is interesting to note the 500/million and 4000/million. Are these health researchers (ie in line with the assertions of the paper?) or researchers in all disciplines? If the latter then it would be helpful for the context to know the former. Is there an ideal ratio that countries should be striving towards? Are there other indicators beyond numbers of researchers/million that provide a sense of the country’s capacity to conduct research? (ie there may be few researchers but very highly competent and producing the same amount of publications as countries with higher proportions. Or few researchers but having high policy impact vs many researchers having high publication impact only). Placing a bit of nuance into the ratios would strengthen the comparisons and the argument being made (if this information is available).
- Line 40. Need a period after “persist”
- Throughout: would be good to use acronym LMICs after first use of the long form.
- Purpose: Line 28: “benchmarking against an “optimal” scenario makes sense but how is “optimal” decided? I assume that box 1 provide references to the literature in

terms of what should be considered as optimal and this has been collated into a list of indicators and components seen in box 2 but this is not clear in the intro or methods sections.

Methods

- Line 41: The MCDC website notes work in 8 African countries. But the paper doesn't clarify whether the 4 African universities that the authors focused on were all in one country? In East and West Africa? South? While it may be important to maintain anonymity of the unis more about their context (geography, size, maturity as mentioned) and the variety would be helpful. The current details are rather vague. Perhaps a table of the four universities would be helpful including Region, Size, Year established etc. If indeed this table would very easily identify the various universities then a more specific narrative would be helpful. How many were west Africa? How many from East etc? Are there within region comparisons?
- Page 5: Line 15 – The authors refer to websites and documents but none are referenced.
- Page 5- line 17 – Its not clear if these “researchers, grants managers and research finance officers” were from the 4 african unis? From the agencies whose websites were availed? From the authors’ institutions? It seems a little clearer later in the paper but would help to have this articulated when first introduced.
- Page 5- line 43. Sentence is incomplete.
- Page 7 – Line 20: What determined “good” or little progress? Was it a minimum score on all indicators? Was it quantified? A little bit more detail on the objectivity of this would be helpful since “subjectively” as a reader I would rate some of these indicators quite differently from what the authors have proposed.
- Was there ethics approval for this study? If so by which institutions?

Results

- page 8: While I appreciate the importance of maintaining anonymity of respondents it would be important to display the diversity of your respondents by labeling them as “university leadership, institution 1”, “staff respondent #11, institution 4” etc...(similar to the labeling in table 2) The way that quotes are currently displayed makes it difficult to appreciate whether these are quotes from one person? From one institution? Etc...
- page 8: were there any negative results? Any learnings from the process (besides the language concerns)?
- Table 2: the particulars per institution is extremely interesting to see in this way. It is surprising however that there is a general statement about progress for each component across the 4 institutions. From the content it would seem that within each component, the level of

progress per institution varies. For instance “research strategies and policies” it would seem like institution 2 has actually made moderate progress while institution 3 made little progress. This seems apparent across the table. This comes back to the comment under methods about how “progress” was measured. Perhaps the level of progress should be indicated under each component for each institution rather than generally across all – particularly because the indicators within each component (shared in Box 2) are many but table 2 only provides a few. This would intuitively make more sense given that the contexts for all 4 are different and it is likely that progress in these various areas is influenced by these particular contexts.

- Table 2: It would be helpful to know what the before and after status on this components was for each institution in order to better understand progress on the said indicators. For instance, the fact that there is no mention of upgrading lab resources in institution 1 could be because they already have this well in place or that there is no movement on it. Without the pre-post, the reader can't get a sense of how progress is being determined.
- Table 2: Under “institution 3, HR management component” cell: “Post- doctoral programmes are not yet institutionalized” and Hr development “More work needs to be done”. These are instances in the table where “non-progress” is mentioned. While this is important it seems odd to classify it under “moderate progress”. This goes back to what the baseline showed versus the post-intervention for the various institutions.
- It may require changing the title of the table as well if statements of needed improvements are also being included.
- Box 1: The Africa Hub produced an assessment tool for research capacity in African universities as well. Wondering if this was perused? The article seems to have been referenced (Jessani et al: Institutional capacity for health systems research in East and Central African schools of public health: experiences with a capacity assessment tool) but the tool wasn't (12961_2013_323_MOESM1_ESM.docx). To what extent were some of the indicators similar or useful? Similarly it would helpful to know what aspects of the resources in box 1 were extracted into the final assessment tool used by the authors.
- Perhaps Box 1 could be replaced with a table that includes the main components, the references, and comments/reflections on each. A short explanation of why this combination was considered “optimal” would provide some insight for the readers as well.
- Else perhaps box 1 and box 2 could be combined to provide the indicator, the reference document that recommends it etc...

Contributor statement

- While the authors have been diligent about protecting the identity of the 4 universities, readers can surmise which ones they were through the authors' contributions (particularly those for data collection). Since this is the case,

	it may strengthen the paper to mention the countries and the universities within the paper anyway without pointing any results directly at any one institution. At the moment the vagueness of “4 African universities” is more of a hindrance since the reader can’t understand context in the least.
--	---

VERSION 1 – AUTHOR RESPONSE

Reviewer: 1

Is the abstract accurate, balanced and complete?

The conclusion of the abstract presents some result points. For instance « Feedback indicated that the process was comprehensive and generated practical actions, several of which were no-cost.” , belongs to the results paragraph rather than to the conclusion. I encourage the authors to review the whole abstract, to check similar inconsistencies. This should help to have a more balanced abstract.

3. We have made adjustments to several sections of the abstract including the changes suggested by the reviewer

Are the methods described sufficiently to allow the study to be repeated?

In the current state of the manuscript, the methods are not appropriately reported.

The study design should be clearly defined and justified with regard to the study purposes; thus the link with the data collect technics, tools and analysis can be better understood.

4. A section has been included in the introduction which outlines the approach to the study and make an explicit link between the purposes and the data collection and analysis processes

Authors conducted a literature review by combining different procedures including internet search. They missed to present useful details about this review. For instance for the internet search : what databases have been reviewed ? when ? which key-words ? What were the selection criteria etc. Other questions include: who developed the tools ? who collected the baseline data ? who collected the follow-up data ? Same or different researchers ? What were their background ?

5. Detailed information about the literature search and resources are provided in the supplementary file. The authors’ ‘contributorship statement’ section has been expanded to cover the topics listed by the reviewer.

Are research ethics (e.g. participant consent, ethics approval) addressed appropriately?

The manuscript do not present the ethical considerations related to this research.

6. At the end of the methods we have added a section on ethics. This is a summary of the more detailed information provided to the journal editors which is copied in full below.

This study was an evaluation of research systems designed to improve practices for strengthening research capacity in universities in LMICs. We followed standard practice for all good quality evaluations which is to obtain information by a variety of methods including interviews with those directly involved in the project being evaluated and with those who may be impacted by the project. Prior to this study we had discussions with various members of the LSTM ethics committee about whether ethical approval was needed for such an evaluation and we sought advice from people whose organisations fund and conduct evaluations. The overall opinion was that ethical approval is not required for evaluations, and this is supported by, for example, WHO* who state that the primary

intent of evaluation (for which ethical approval is not needed) is to “improve a public health program or service’. We therefore did not seek formal ethical approval for this study. However we did explain the study to participants, we asked each interviewee for their verbal consent to participate and we provided an opportunity for them to refuse without any consequences for themselves.

We recognise that this is a grey area between research and evaluation, and this project prompted further discussions with our ethics committee. Together we have devised a process for subsequent studies of this nature which involves informing our ethics committee and providing all those involved in the evaluation with information about the purpose and methods, and issuing each institution with a consent form.

*World Health Organisation (WHO), 2010. Research Ethics in International Epidemic Response; WHO Technical Consultation (pdf). Geneva: WHO. Available online at: http://www.who.int/ethics/gip_research_ethics_.pdf

Are the results presented clearly?

The structure of the results should be linked to the study design, that seems to be a multiple case study design. If this suggested design is accepted by authors, the results will gain to identify similarities and differences rather than providing only a listing of separate findings in each case (institution). Beyond what can be learned in each cases, authors should analyse what can be learned across cases. This reference may help : Yin R: Case study research. Design and methods. Los Angeles: Sage Publications; 2009.

7. Commonalities across the institutions (i.e. they occurred in at least three of the four universities) and proposed actions that emerged during the on-site visits to address these gaps, are already summarised in table 2. We also compared progress in addressing capacity gaps among the institutions (see table 3 column one).

Are the study limitations discussed adequately?

Overall the study focuses on gaps and ignore explicitation of existing capacities, and existing good practices. The paper focuses also on the meso-level (academic organization). Whoever several meso-level factors are obviously influenced by the macro-level factors like the National Health Research System (Tikki P, Ritu s, Sadana SH, Zulfiqar AB, A.H. A, Jonathon S. Knowledge for better health — a conceptual framework and foundation for health research systems. Bulletin of World Health Organization. 2003;81:6). The discussion may gain to discuss this interlinks and the choice made by the authors.

8. We have included many examples of innovative practice and problem-solving in table 3 (last column) and this table also highlights many of the positive changes in RMSS capacity that have occurred in the institutions. The study does focus on capacity gaps since the purpose was to identify gaps in institutions” RMSS so the institutions could put measures in place to address them.

Reviewer: 2

Overall comments:

- The paper refers to health research in a more general term throughout the document but it would seem like the authors are referring particularly to policy relevant or operational health research (as opposed to medical health research). If this is indeed the correct interpretation then it would be helpful to make this more explicit.

9. We have changed ‘health research’ to ‘operational health research’ where appropriate throughout

the paper

- The identity of the 4 institutions can be surmised from the author contributions. Given that the paper would be stronger if the list of institutions and countries was included in the intro and methods sections of the paper while maintaining anonymity in the results (see comments in the abstract, intro and methods section). This would provide a better sense of context (4 african countries out of 54 underappreciates the diversity and complexity within the continent (and oftentimes within a country too).

10. In the methods we have included information about the regional location of the institutions within Africa without naming them explicitly. We have also added a statement in the limitations reflecting the reviewer's bracketed statement

- The details on how "progress" is being evaluated is extremely vague. The nebulousness creates confusion and frustration at various areas of the paper and it would be helpful to have this articulated with a bit more detail and with the pre-post status of these institutions with respect to the various indicators (see comments in results section)

11. See responses 26, 29, 30, 21 for specific information about how we have addressed this comment

Specific comments:

Abstract:

- There seems to be a background section missing...This would perhaps link to the find sentence in the conclusions about investing in research capacity.

12. We have made adjustments to several sections of the abstract including adding an introductory sentence providing background to the study

- It would be helpful to break up the participants into how many were staff and how many were students.

13. A new table showing a breakdown of participants by institution and cadre (table 1) has been added and referenced in the results section

- Were the 4 African universities all in one country? In East and West Africa? South? While it may be important to maintain anonymity of the unis more about their context would be helpful under Setting or include it in Intervention/methods.

14. Please see response 10

- Mixed-methods research often implies quant as well as qual methods. Might be more informative for the reader if the authors indicated "qualitative methods" and then listed the three.

15. We have made changes to several parts of the abstract including clarifying the methods used in the study

- "progress under indicators was tracked" implied a timeframe that is not included in the abstract. This would be helpful too.

16. The time period (15 months) is stated in the methods but is now also in the abstract

Introduction

- Line 20: "Few African countries" is vague. Would be helpful to know how many and which ones.

17. We have now included information about these countries in the paper.

- Line 32: It is interesting to note the 500/million and 4000/million. Are these health researchers (ie in line with the assertions of the paper?) or researchers in all disciplines? If the latter then it would be helpful for the context to know the former. Is there an ideal ratio that countries should be striving towards? Are there other indicators beyond numbers of researchers/million that provide a sense of the country's capacity to conduct research? (ie there may be few researchers but very highly competent and producing the same amount of publications as countries with higher proportions. Or few researchers but having high policy impact vs many researchers having high publication impact only). Placing a bit of nuance into the ratios would strengthen the comparisons and the argument being made (if this information is available).

18. The numbers refer to researchers of all disciplines and this has been clarified in the paper. We have searched for information about more nuanced measures of research capacity but have only been able to find similar quantitative targets and data (e.g. http://www.unesco.org/new/en/media-services/single-view/news/how_much_do_countries_invest_in_rd_new_unesco_data_tool_re/)

- Line 40. Need a period after "persist"

19. Done

- Throughout: would be good to use acronym LMICs after first use of the long form.

20. Done

- Purpose: Line 28: "benchmarking against an "optimal" scenario makes sense but how is "optimal" decided? I assume that box 1 provide references to the literature in terms of what should be considered as optimal and this has been collated into a list of indicators and components seen in box 2 but this is not clear in the intro or methods sections.

21. The reviewer has accurately summarised what we did. We have provided more explanation in the introduction and the methods about how 'optimal' was decided and the relationships between the literature and the final list of indicators and components. The 'optimal' scenario for an institutional RMSS was derived by amalgamating all the items identified from the literature search and eliminating any redundancy. In order to ensure comprehensiveness and minimise bias, no assumptions were made about what should be included, no selection criteria were applied to the original list of items and they were drawn together under the eight components without losing any of the items.

Methods

- Line 41: The MCDC website notes work in 8 African countries. But the paper doesn't clarify whether the 4 African universities that the authors focused on were all in one country? In East and West Africa? South? While it may be important to maintain anonymity of the unis more about their context (geography, size, maturity as mentioned) and the variety would be helpful. The current details are rather vague. Perhaps a table of the four universities would be helpful including Region, Size, Year established etc. If indeed this table would very easily identify the various universities then a more specific narrative would be helpful ie how many were west Africa? How many from East etc? Are there within region comparisons?

22. Please see response 10. Anonymised comparisons are provided in table 2

- Page 5: Line 15 – The authors refer to websites and documents but none are referenced.

23. these are listed in Supplementary file Box 1

- Page 5- line 17 – Its not clear if these “researchers, grants managers and research finance officers” were from the 4 african unis? From the agencies whose websites were availed? From the authors’ institutions? It seems a little clearer later in the paper but would help to have this articulated when first introduced.

24. This has been clarified in the text as ‘within and beyond our own institutions’

- Page 5- line 43. Sentence is incomplete.

25. ‘determined’ has been added to complete the sentence

- Page 7 – Line 20: What determined “good” or little progress? Was it a minimum score on all indicators? Was it quantified? A little bit more detail on the objectivity of this would be helpful since “subjectively” as a reader I would rate some of these indicators quite differently from what the authors have proposed.

26. We considered, but rejected, the idea of having numerical scores to represent progress since gauging progress was subjective. Instead, progress was self-rated by the principal investigators in discussion with authors SW and IB. This had already been included as a study limitation.

- Was there ethics approval for this study? If so by which institutions?

27. Please see response 6

Results

- page 8: While I appreciate the importance of maintaining anonymity of respondents it would be important to display the diversity of your respondents by labeling them as “university leadership, institution 1”, “staff respondent #11, institution 4” etc...(similar to the labeling in table 2) The way that quotes are currently displayed makes it difficult to appreciate whether these are quotes from one person? From one institution? Etc...

- page 8: were there any negative results? Any learnings from the process (besides the language concerns)?

28. The quotes are all from principal investigators. We explain this in the methods section of the paper: ‘These comments were organized into themes, and quotes reflective of each theme were selected to convey the principal investigators’ perspectives in their own words.’ We have also now explained this in the sentence preceding the quotes. Examples of the learnings are described in the last column of table 2 (termed ‘examples of innovation and problem-solving’), in the discussion and in the ‘strengths and limitations’ section.

- Table 2: the particulars per institution is extremely interesting to see in this way. It is surprising however that there is a general statement about progress for each component across the 4 institutions. From the content it would seem that within each component, the level of progress per institution varies. For instance “research strategies and policies” it would seem like institution 2 has actually made moderate progress while institution 3 made

little progress. This seems apparent across the table. This comes back to the comment under methods about how “progress” was measured. Perhaps the level of progress should be indicated under each component for each institution rather than generally across all – particularly because the indicators within each component (shared in Box 2) are many but table 2 only provides a few. This would intuitively make more sense given that the contexts for all 4 are different and it is likely that progress in these various areas is influenced by these particular contexts.

29. The reviewer is correct to point out that progress per institution varied with different components. We know that some of the institutions used study findings (provided in detail in institution-specific reports) to develop institutional capacity strengthening plans targeting RMSS gaps identified through this study. However, rather than present an in-depth analysis of each institution we preferred to focus on the higher-level lessons around what aspects of RMSS capacity strengthening the African institutions commonly found most easy and most difficult to address. Our reasons for this were twofold. Firstly we considered that journal readers would find it more useful to understand the higher-level lessons emerging from this study rather than the details of progress within individual institutions since this information was likely to be more generalizable and transferable. Secondly, governments and external funding agencies, with whom we work closely, are currently very interested in knowing where and how to target their capacity strengthening efforts and resources. Our higher-level findings are therefore of great relevance to them since they need to invest at national and supra-national, rather than institutional, level. All this is explained in the ‘implications’ section of the paper

- Table 2: It would be helpful to know what the before and after status on this components was for each institution in order to better understand progress on the said indicators. For instance, the fact that there is no mention of upgrading lab resources in institution 1 could be because they already have this well in place or that there is no movement on it. Without the pre-post, the reader can't get a sense of how progress is being determined.

30. We have added details into table 2 to better explain changes from baseline status where this was not clear (for example, indicating when initiatives are new or continuations/ongoing). The table is not designed to comprehensively show all activities across every item within each of the eight components. Rather, it provides ‘Examples of innovative practices and problem-solving’ for various RMSS components. The table is therefore illustrative to demonstrate how such information can be captured and communicated and we have altered the table title to make this explicit and further emphasised it in the results section.

- Table 2: Under “institution 3, HR management component” cell: “Post- doctoral programmes are not yet institutionalized” and Hr development “More work needs to be done”. These are instances in the table where “non-progress” is mentioned. While this is important it seems odd to classify it under “moderate progress”. This goes back to what the baseline showed versus the post-intervention for the various institutions.

- It may require changing the title of the table as well if statements of needed improvements are also being included.

31. While we agree that the examples provided by the reviewer do demonstrate little/no progress, the classification ‘moderate progress’ refers to overall progress on all activities within this theme across all institutions (see heading of column 1, table 2). We feel that the table title ‘ Progress on actions.....’ implies that ongoing improvements are still needed without needing to change the table heading

- Box 1: The Africa Hub produced an assessment tool for research capacity in African universities as well. Wondering if this was perused? The article seems to have been referenced (Jessani et al: Institutional capacity for health systems research in East and

Central African schools of public health: experiences with a capacity assessment tool) but the tool wasn't (12961_2013_323_MOESM1_ESM.docx). To what extent were some of the indicators similar or useful? Similarly it would be helpful to know what aspects of the resources in box 1 were extracted into the final assessment tool used by the authors.

- Perhaps Box 1 could be replaced with a table that includes the main components, the references, and comments/reflections on each. A short explanation of why this combination was considered "optimal" would provide some insight for the readers as well.
- Else perhaps box 1 and box 2 could be combined to provide the indicator, the reference document that recommends it etc...

32. Our literature search to inform the design of this project preceded the publication of the Jessani et al 2014 paper. However we were aware of this work, and the capacity assessment tool, when writing up the study and therefore referenced this publication (which includes the capacity assessment tool, in the paper). Many of the indicators in the capacity assessment tool in the Additional File 1 in the Jessani reference (e.g. strategies, collaborations) are similar to those that we used, though in the Jessani there is more emphasis on (semi) quantitative data whereas we used more qualitative methods.

See response 21 for a description of what aspects of the resources in Box 1 (referred to as 'items' in our paper) were extracted into the final assessment tool and for information about how we determined the 'optimal' scenario.

We have considered the reviewer's suggestion of combining boxes 1 and 2 or replacing box 1 with a table of all the components/items and giving a reflection on each. However, there was much duplication in the items that we extracted from the literature so producing a table of all the items from each of the resources would result in an unwieldy and unfocused table.

Contributor statement

- While the authors have been diligent about protecting the identity of the 4 universities, readers can surmise which ones they were through the authors' contributions (particularly those for data collection). Since this is the case, it may strengthen the paper to mention the countries and the universities within the paper anyway without pointing any results directly at any one institution. At the moment the vagueness of "4 African universities" is more of a hindrance since the reader can't understand context in the least

33. Please see response 10

VERSION 2 – REVIEW

REVIEWER	Dossou Jean-Paul Centre de Recherche en Reproduction Humaine et en Démographie, Cotonou, Benin & Institute of Tropical Medicine of Antwerp, Belgium
REVIEW RETURNED	17-May-2017

GENERAL COMMENTS	Comment 1: This version is better. I thank the authors for their efforts. Thank you. Comment 2: Methods: Overall this study used qualitative data collection techniques and tools (interview guides, document reviews and facilities observation guides). The authors didn't report any quantitative analysis of the qualitative data collected. Is this study still a mixed method study? Comment 3: Ethics: I acknowledge the reference to the grey zone between evaluation and research. It is good that authors engaged in further discussion with their ethics committee. I would like however to share the following analysis, that may help in this discussion. The primary intent of this project was to "improve a public health program
--

	or service”. Even in this case, risks for participants need to be assessed and addressed properly. First, there are for instance a moral risks for individual participants at a lower level in the academic hierarchy, as they may suffer from administrative or other harassments if they share negative comments about the university performance or about the leadership capacity of some specific actors in the university. Authors mentioned that they gave “an opportunity for them (participants) to refuse without any consequences for themselves”, but this is unsificiant for instance to protect participants against internal harrasements following the study. A key point in this case is for instance the respect of the confidentiality, and active measures to avoid the possibility to link data to the participants. A proper integration of this concern in the process may even improve the confidence of the participants and improve the quality of the data collected. Second, this “evaluation” could have remained internal to the organizations involved or be disseminated only at a small local level. But the publication toward an international audience creates different ethical concerns. For instance, the university as a moral entity may gain further credibility but also because of the publication of the findings, those universities, may loose some credibility and or some resources. Do you share this analysis? Are those concerns relevant from your point of view? Did you anticipate about those concerns? How did you minimize them beyond the oral information and the verbal consent?
--	---

REVIEWER	Nasreen Jessani Johns Hopkins University, USA. Stellenbosch University, South Africa
REVIEW RETURNED	25-May-2017

GENERAL COMMENTS	The authors have comprehensively and adequately responded to all previous queries. Thank you! There are only a few minor comments. Minor Comments: Overall: The manuscript alternates between referring to the 4 sites as “universities” and “institutions”. Would be better to choose one for consistency. Suggest “universities” as it is more specific and relevant to the paper. Title: Should it read “processes and tools” or “a process and tool”? A bit strange to read as is. Abstract: It's a little ambitious to state “assessing African universities’....” when there are 54 countries in Africa and over 600 accredited universities. Kindly specify that this is for only 4 universities in Africa. This will align well with the acknowledgement in the limitations section. Introduction Line 24: Lack of research/researchers in LMICs especially Africa. Since Africa is not an LMIC (country), I would suggest including rephrasing to “... especially on the African continent.” Or include “in” before “Africa.” Purpose: Line 18: include the number 4 before African Universities to parallel
--

	the abstract Methods: Page 5 line 9: the locations were nicely introduced on line 4 (thank you!) so would suggest deleting the redundancy in line 9. Results: Authors note that interviews were with 11 cadres of respondents yet the quotes are derived from only one cadre (the PIs)? Is the readership to assume that the voices of others have been captured in the summarized content of the results with none necessary to highlight?
--	--

VERSION 2 – AUTHOR RESPONSE

Reviewer: 1

Reviewer Name: Dossou Jean-Paul

Comment 1: This version is better. I thank the authors for their efforts. Thank you.

Comment 2: Methods: Overall this study used qualitative data collect technics and tools (interview guides, document reviews and facilities observation guides). The authors didn't report any quantitative analysis of the qualitative data collected. Is this study still a mixed method study?

>> We agree that the data we present is primarily qualitative and have altered 'mixed methods' to 'qualitative' where appropriate throughout the paper.

Comment 3: Ethics: I acknowledge the reference to the grey zone between evaluation and research. It is good that authors engaged in further discussion with their ethics committee. I would like however to share the following analysis, that may help in this discussion. The primary intent of this project was to "improve a public health program or service". Even in this case, risks for participants need to be assessed and addressed properly. First, there are for instance a moral risks for individual participants at a lower level in the academic hierarchy, as they may suffer from administrative or other harassments if they share negative comments about the university performance or about the leadership capacity of some specific actors in the university. Authors mentioned that they gave "an opportunity for them (participants) to refuse without any consequences for themselves", but this is unsufficient for instance to protect participants against internal harrasements following the study. A key point in this case is for instance the respect of the confidentiality, and active measures to avoid the possibility to link data to the participants. A proper integration of this concern in the process may even improve the confidence of the participants and improve the quality of the data collected. Second, this "evaluation" could have remained internal to the organizations involved or be disseminated only at a small local level. But the publication toward an international audience creates different ethical concerns. For instance, the university as a moral entity may gain further credibility but also because of the publication of the findings, those universities, may loose some credibility and or some resources. Do you share this analysis? Are those concerns relevant from your point of view? Did you anticipate about those concerns? How did you minimize them beyond the oral information and the verbal consent?

>> We thank the reviewer for these careful considerations about the ethics of this type of research. We fully recognise the issues that the reviewer has raised and have had extensive discussions about them with various ethics committees and funding bodies. In an effort to get some coherent guidance about these trans-national ethical issues we are currently in discussion with the Wellcome Trust, the African Academy of Science and others about how to approach such challenges in the future. We are actively pursuing the possibility of conducting research on this issue to collate and expand existing evidence to inform the discussions.

Regarding minimising risks, we did take several actions in addition to consent. For example no internal person from the university was present at the interviews and all reports (even internal ones) were anonymised so no individuals apart from the main collaborator were named. Through the main collaborator all those who were interviewed were invited to participate in shaping the report and the report was shared with key stakeholders in the institution, but not beyond them. The collaborators were aware from the beginning of the project that there was a possibility that the results would be published (anonymously) jointly with the key players and they were content to proceed on that basis.

Reviewer: 2

Reviewer Name: Nasreen Jessani

Overall:

The manuscript alternates between referring to the 4 sites as “universities” and “institutions”. Would be better to choose one for consistency. Suggest “universities” as it is more specific and relevant to the paper.

>> some of the organisations were universities but others were research institutes which did not regard themselves as universities. We have made this clearer in the first line of the methods (‘We worked with four African universities or research institutions’). Rather than use both the terms ‘universities’ and ‘institutes’ throughout the paper, which would be too cumbersome, we have generally kept to the more general term ‘institutions’.

Title: Should it read “processes and tools” or “a process and tool”? A bit strange to read as is.

>> the title has been changed to ‘processes and tools’ since we used three different tools, each with its own process.

Abstract: It's a little ambitious to state “assessing African universities’....” when there are 54 countries in Africa and over 600 accredited universities. Kindly specify that this is for only 4 universities in Africa. This will align well with the acknowledgement in the limitations section.

>> although we have already stated ‘four’ in the ‘setting’ section of the abstract we have now also changed the wording in the objectives section

Introduction

Line 24: Lack of research/researchers in LMICs especially Africa. Since Africa is not an LMIC (country), I would suggest including rephrasing to “... especially on the African continent.” Or include “in” before “Africa.”

>>changed to ‘especially in Africa’

Purpose:

Line 18: include the number 4 before African Universities to parallel the abstract

>> done

Methods:

Page 5 line 9: the locations were nicely introduced on line 4 (thank you!) so would suggest deleting the redundancy in line 9.

>> we have deleted the redundant statement in line 9

Results:

Authors note that interviews were with 11 cadres of respondents yet the quotes are derived from only one cadre (the PIs)? Is the readership to assume that the voices of others have been captured in the summarized content of the results with none necessary to highlight?

>> baseline data were collected from the 11 different cadres; follow up data (to which the quotes

refer) were collected from only the PIs. We have made this clearer in the description of how baseline data were collected (p6).

VERSION 3 – REVIEW

REVIEWER	Nasreen Jessani Johns Hopkins University, USA
REVIEW RETURNED	31-May-2017

GENERAL COMMENTS	Thank you for such an interesting study and paper. It is a great addition to the literature and to practical interventions currently in academia! Wishing the team the best of luck with future studies!
--